# Design and Assessment of a Novel In Silico Approach for Developing a Next-Generation Multi-Epitope Universal Vaccine Targeting Coronaviruses

**DOI:** 10.3390/microorganisms11092282

**Published:** 2023-09-11

**Authors:** Muhammad Asif Rasheed, Sohail Raza, Wadi B. Alonazi, Muhammad Adnan Ashraf, Muhammad Tariq Navid, Irfana Aslam, Muhammad Nasir Iqbal, Sarfraz Ur Rahman, Muhammad Ilyas Riaz

**Affiliations:** 1Department of Biosciences, COMSATS University Islamabad, Sahiwal Campus, Sahiwal 57000, Pakistan; asif.rasheed@cuisahiwal.edu.pk (M.A.R.); irfanaaslam0786@gmail.com (I.A.); nasir.iqbal@iub.edu.pk (M.N.I.); 2College of Veterinary Medicine, Huazhong Agricultural University, Wuhan 430070, China; 3Institute of Microbiology, University of Veterinary and Animal Sciences, Lahore 54000, Pakistan; adnan.ashraf@uvas.edu.pk (M.A.A.); ilyas.riaz@uvas.edu.pk (M.I.R.); 4Health Administration Department, College of Business Administration, King Saud University, Riyadh 11587, Saudi Arabia; 5Department of Biological Sciences, National University of Medical Sciences, Rawalpindi 46000, Pakistan; tariq.navid@numspak.edu.pk; 6Department of Parasitology, University of Veterinary and Animal Sciences, Lahore 54000, Pakistan; sarfrazvet@gmail.com

**Keywords:** SARS-CoV-2, variants, universal vaccine, multi-epitope, prevention, bioinformatics

## Abstract

In the past two decades, there have been three coronavirus outbreaks that have caused significant economic and health crises. Biologists predict that more coronaviruses may emerge in the near future. Therefore, it is crucial to develop preventive vaccines that can effectively combat multiple coronaviruses. In this study, we employed computational approaches to analyze genetically related coronaviruses, including severe acute respiratory syndrome coronavirus 2 (SARS-CoV-2) and its variants, focusing on the spike glycoprotein as a potential vaccine candidate. By predicting common epitopes, we identified the top epitopes and combined them to create a multi-epitope candidate vaccine. The overall quality of the candidate vaccine was validated through in silico analyses, confirming its antigenicity, immunogenicity, and stability. In silico docking and simulation studies suggested a stable interaction between the multi-epitope candidate vaccine and human toll-like receptor 2 (TLR2). In silico codon optimization and cloning were used to further explore the successful expression of the designed candidate vaccine in a prokaryotic expression system. Based on computational analysis, the designed candidate vaccine was found to be stable and non-allergenic in the human body. The efficiency of the multi-epitope vaccine in triggering effective cellular and humoral immune responses was assessed through immune stimulation, demonstrating that the designed candidate vaccine can elicit specific immune responses against multiple coronaviruses. Therefore, it holds promise as a potential candidate vaccine against existing and future coronaviruses.

## 1. Introduction

The coronavirus SARS-CoV-2 originated in Wuhan and rapidly spread across the globe within three months. The number of infected cases reached a staggering 689 million, resulting in over 6.8 million deaths worldwide. SARS-CoV-2 is a novel addition to its group. Based on genome sequences, coronaviruses are categorized into four primary types: alphacoronaviruses, betacoronaviruses, gammacoronaviruses, and deltacoronaviruses [1]. Among these, the betacoronaviruses pose a significant threat to humanity, as manifested in outbreaks at the turn of the century. This category includes the severe acute respiratory syndrome virus (SARS-CoV) and the Middle East respiratory syndrome virus (MERS-CoV), which caused major epidemics in 2003 and 2012, respectively. This marks the third instance of a coronavirus affecting the world, presenting a heightened level of danger and an alarming situation.

The coronaviruses are enveloped viruses having a spherical or pleomorphic shape and contain positive-sense RNA. The genome ranges from 26 to 32 Kbps with 80–120 nm of diameter. The viral genome contains structural proteins; namely, envelope protein (E), membrane protein (M), nucleo-capsid protein (N), and spike protein (S). The E-protein is involved in viral budding and releases while the M-protein helps in virus assembly and budding. The nucleo-capsid protein is used for binding and viral RNA packaging in the assembly of the virion [1,2]. The S-protein (also called surface protein) involves viral attachment through the receptor(s) (ACE2 and others) to host cells and acts as the main target protein for neutralizing antibodies. This is also a target for antiviral peptides [3].

To control influenza viral infections, a “universal vaccine” has been introduced to produce long-lasting immunity against all serotypes. The two surface proteins based on which the influenza virus is classified are hemagglutinin (H) and neuraminidase (N). There are eighteen H subtypes and eleven N subtypes. The H-protein (HA) is responsible for viral entry into a human cell. This protein is used in most of the vaccine development strategies. The HA has two parts, the head and the stem. In the vaccine, the HA-head is used as a target to induce antibodies. Recently, scientists have found that the head of the HA is a variable component of H-protein while the stem remains unchanged. Thus, the HA-stem has been suggested as an ideal target for the development of a universal vaccine [4].

The RNA viruses generally perform modifications in their genetic make-up (sequences) through mutation that results in the development of new variants. Due to this strategy, these viruses are predicted to cause the worst global pandemics. These evolutions and genetic changes have been observed in influenza and coronaviruses. The SARS-CoV and SARS-CoV-2 viruses shared high genetic similarities [5].

The S-protein of the coronaviruses (CoVs) is an immunogenic protein that has been used in various studies to induce neutralizing antibodies. The S-protein has been divided into an S1 subunit that has a receptor-binding domain (RBD) that attaches to a cellular receptor (angiotensin-converting enzyme 2 (ACE2), and an S2 subunit that is used to facilitate fusion between a virus and susceptible cell to release viral RNA into the cytoplasm and for replication [6]. In multiple vaccine development strategies, the S-protein is used as a full-length component, or only the S1 region’s RBD, or as expressed in virus-like particle (VLP) or viral vectors [7,8]. The C-terminus in the S1 subunit region has been expressed as an immune-dominant portion in the *Deltacoronaviruses* that can potentially induce higher neutralizing antibodies [9,10]. These results highlighted the importance of S-protein (as its full length or its immunogenic portion(s)) as a promising candidate for provoking virus-neutralizing antibodies and, hence, for uplifting T-cell responses. The S-protein was found to be an excellent target for humoral immune responses. The duration of neutralizing antibodies has been found to be shorter in recovered patients as compared to T-cell responses [11,12,13]. The T-cell responses against a structural protein were more immunogenic than against nonstructural protein, while the cellular immune responses against S- and N-proteins were found to be higher and longer-lasting than that of other proteins [14,15]. Hence, collectively, these findings highlight the importance of T-cell responses and should attract scientists’ attention to develop novel vaccines. 

The genomic studies of coronaviruses showed very similar genetic makeups [5]. The SARS-CoV-2 variants’ emergence, especially that of variants having critical mutations in the RBD of the spike protein, has increased the global spread of COVID-19. Many of these variants have the ability to spread more easily and have more pathogenic potential as compared to other variants [16]. Some of these variants are able to produce reinfections via reducing the sensitivity of neutralizing antibodies produced by the first-generation vaccines or COVID-19 infection [17,18]. There is an opinion that we need a universal SRA-CoV-2 vaccine regardless of antigenic drift, antigenic shift, and virus subtypes. This type of universal vaccine will able to provide protection against currently circulating and new emerging SARS-CoV-2 subtypes [19]. 

Modern vaccine development strategies utilize immuno-informatics for the designing of novel and potentially active vaccine(s). This technology has been utilized effectively in the identification of cytotoxic T lymphocyte cells (CTL) and B-cell epitopes in the S-protein of SARS-CoV-2. These epitopes do interact with their corresponding MHC-I molecules and generate immune responses. The CLT epitopes bind multiple times with MHC-I peptide-binding grooves, which were studied by using molecular dynamic simulations [20]. These multiple binding epitopes provided an excellent opportunity that needs to be observed for the development of a universal vaccine. Recently, similar strategies were used to develop multi-epitope vaccines using in silico approaches and effective immune responses in vivo were produced [21,22,23]. In these vaccines, a limited number of SARS-CoV-2 sequences were used, but the coronavirus is changing as the number of infections increases. Therefore, there is a dire need to use more sequences and develop vaccines that can be beneficial for new variants and future coronavirus outbreaks. This candidate vaccine will be a very effective candidate vaccine in inducing neutralizing antibodies that will be long-lasting and equally effective, not only for currently present SARS-CoV-2, but also for future variants, as the coronaviruses will repeat their outbreaks in the next few years.

## 2. Materials and Methods

### 2.1. Retrieval of Sequence

The flowchart of the work is shown in Figure 1. The protein sequences of SARS-CoV-2 were retrieved from the NCBI protein database (https://www.ncbi.nlm.nih.gov/protein/, accessed on 28 February 2022) (National Library of Medicine, Bethesda, MD, USA). Altogether, 175 sequences for SARS-CoV-2 were retrieved, including 24, 24, 24, 17, 24, 24, 12, 24 and 2 sequences of Alpha, Beta, Delta, Eta, Gamma, Keppa, Mu, Omicron and Zeta, respectively. All sequences were related to a surface glycoprotein of the coronaviruses. 

### 2.2. Generation of a Consensus Sequence

Geneious Prime software version 2020.1.2 was used to generate a consensus sequence [24]. The software first performs the alignment of sequences, then generates the consensuses sequence from the aligned sequences. This software is a very useful and comprehensive tool for researchers in the field of molecular biology. Moreover, the tool is also used for the analysis of next-generation sequencing (NGS) sequence data. Hence, due to its usage, the tool is one of the leading bioinformatics software platforms. The sequences were uploaded manually on the Geneious Prime tool. The software was run using default parameters. Moreover, the BLOSUM 62 matrix was used to calculate the alignment score. The aligning and assembling of multiple sequences were performed using the without gaps option.

### 2.3. The Prediction of Epitopes and Their Joining

The prediction of epitopes constitutes a pivotal phase in the development of multi-epitope vaccines. To this end, diverse categories of epitopes affiliated with cytotoxic T lymphocyte (CTL) and helper T lymphocyte (HTL) subsets were forecasted. The epitopes were predicted utilizing the NetCTL 1.2 webserver (https://www.cbs.dtu.dk/services/NetCTL/, accessed on 5 March 2022) (Technical University of Denmark, Lyngby, Denmark), applying default threshold criteria: 0.15 for proteasomal C-terminal cleavage, 0.05 for Transporter Associated with Antigen Processing, and 0.75 for epitope recognition. The prognostication by NetCTL manifests a sensitivity of 54–89% and specificity of 94–99%. Additionally, the Immune Epitope Consensus (IEDB) tool v2.26 (https://tools.iedb.org/mhci/, accessed on 28 March 2022) was employed for epitope prediction [25]. Following epitope prediction, a sorting process was enacted to eliminate redundant epitopes, subsequently organized based on positional attributes. Moreover, the anticipation of B-cell epitopes was accomplished via the IEDB webserver v2.26 (http://tools.iedb.org/bcell/, accessed on 31 March 2022), recognizing the significance of these epitopes in fostering defensive immunity [26]. The prediction of B-cell epitopes leveraged Hidden Markov Models (HMMs) in conjunction with parameters such as predicted score, antigenicity, allergenicity, and toxicity. HTL epitopes of length 9-mer were projected through the IEDB MHCI tool v2.26 (http://tools.iedb.org/mhci/, accessed on 20 April 2022) using default parameters. All projected epitopes were initially sieved based on default criteria. The chosen epitopes underwent a refinement process, culminating in the selection of antigenic, non-toxic, and non-allergenic epitopes for subsequent scrutiny. Furthermore, redundancy was eradicated, and linkers including EAAAK, AAY, and GPGPG spacers were employed to interlink the epitopes [27]. 

### 2.4. Antigenicity of the Single Peptide Epitope

The ANTIGENpro tool 1.0 (http://scratch.proteomics.ics.uci.edu/, accessed on 27 April 2022) was employed to prognosticate the antigenicity of an individual peptide epitope. This webserver operates on sequence-based principles and is characterized as an alignment-independent mechanism. Notably, its predictive capacity pertains to antigenicity, a feature detached from pathogenic considerations. The predictive process involves the utilization of five distinct machine learning algorithms. Subsequently, the support vector machine (SVM) undertakes the classification of the queried peptide, thus rendering a determination of the protein’s antigenic nature. This process culminates in the derivation of an overarching probability denoting the antigenicity likelihood of the queried peptide. In parallel, the Vaxijen webserver v2.0 (http://www.ddg-pharmfac.net/vexijen/VaxiJen/VaxiJen.html, accessed on 3 May 2022) was also harnessed for analogous objectives, aiming to corroborate the findings. This alternate tool’s predictive approach hinges upon the physicochemical attributes inherent to the provided protein.

### 2.5. Allergenicity Prediction

Allergenicity prediction was performed by using AllergenFP version 1.0 (http://ddg-pharmfac.net/AllergenFP/, accessed on 10 May 2022) and AllerTOP version 2.0 (https://www.ddg-pharmfac.net/AllerTOP/, accessed on 10 May 2022). During the analysis, the program described the dataset by five E-descriptors. Moreover, the strings were converted into vectors. This conversion was done by auto-cross covariance [28]. These E-descriptors have been described already by Venkatarajan and Braun for twenty amino acids. Moreover, principal component analysis (PCA) was used for the prediction. Hence, five principal components (E1–E5) were used. Among these five, E1 is related to the hydrophobicity of amino acids. E2 is related to the size, while E3 is related to the helix propensity of the amino acids. E4 is related to relative abundance, while E5 is related to the beta-strand-forming propensity of the amino acids. During the analysis, Tanimoto coefficients were calculated for the proteins. At the end of the analysis, the peptide was classified as allergen or non-allergen based on the highest Tanimoto coefficient.

### 2.6. Solubility Check by SOLpro

SOLpro v1.0 (http://scratch.proteomics.ics.uci.edu/, accessed on 15 May 2022) was used to predict the solubility of the protein upon over-expression in *E. coli*. This prediction was based on the SVM. Hence, representations of the query sequence were done by different classifiers and the final SVM classifier précises the prediction of the peptide. Based on the analysis, the protein was oriented as soluble or not, as well as the corresponding probability of predicted solubility upon over-expression.

### 2.7. Properties Calculation Using the EXPASY ProtParam Tool

The utilization of the ProtParam webserver v1.0 (https://web.expasy.org/protparam/, accessed on 18 May 2022) facilitated the computation of diverse physicochemical attributes associated with the protein of interest. This computational process relies on an array of distinct physical and chemical parameters cataloged within the Swiss-Prot or TrEMBL databases. The webserver affords the capability to derive a spectrum of properties pertinent to the designated peptides. Illustratively, the server is equipped to evaluate parameters such as the extinction coefficient, instability index, and molecular weight, among others. Furthermore, additional attributes including the grand average of hydropathicity (GRAVY), theoretical isoelectric point (pI), atomic composition, projected half-life, and aliphatic index were also ascertained through this computational framework.

### 2.8. Prediction of TM Helices

The TMHMM version 2.0 (http://www.cbs.dtu.dk/services/TMHMM/, accessed on 19 May 2022) webserver was used to predict the trans-membrane helix of the query protein. This webserver is very good in the context of predicting trans-membrane helices. The query protein is provided as a FASTA sequence to the webserver. After the analysis, the output format can be fetched as long output or short output. Hence, the webserver was used to find out potential trans-membrane helices in the proposed vaccine.

### 2.9. The 3-D Structure Prediction

The 3-D structure of the combined epitope was predicted using the I-Tasser webserver v1.0 (https://zhanggroup.org/I-TASSER/, accessed on 21 May 2022). To predict the structure, initially a homology modeling approach was tried, but due to the unavailability of a suitable template, the homology modeling method was not used and the structure of the protein was predicted through a threading algorithm for protein structure prediction.

### 2.10. Structure Validation by Rampage Analysis

The quality of the structure of the predicted protein was determined by a Ramachandran plot. This is a very useful tool to analyze the structure of the protein. The plot can highlight the interactions that determine the different characteristics of the amino acids [29]. The structure validation of the predicted structure was performed by the Ramachandran Plot server 1.0 (https://www.umassmed.edu/zlab/, accessed on 25 May 2022).

### 2.11. Structure Refinement

After predicting the quality of the structure of the protein, the quality of the structure was further enhanced by using a webserver (http://galaxy.seoklab.org/cgi-bin/report_REFINE.cgi?key=e0867efa82610f5bd5b374be756c152e, accessed on 25 May 2022). The GalaxyREFINE webserver (https://bio.tools/galaxyrefine, accessed on 25 May 2022) was used for this purpose, which predicts an unreliable region of the structure for which the information is not available by the ab initio method. The ab initio loop terminus modeling method is one of the few refinement methods that improves the starting models, as demonstrated in CASP19 [30]. Moreover, the refined structures were further validated using the ProSA-web 1.0 (ProSA-web—Protein Structure Analysis (https://prosa.services.came.sbg.ac.at/prosa.php, accessed on 25 May 2022)) and ERRAT (ERRAT—UCLA-DOE Institute) 1.0 webservers.

### 2.12. Codon Optimization and Reverse Translation and in Silico Cloning

In order to perform the codon optimization and reverse translation, we used the VectorBuilder webserver 1.0 (https://en.vectorbuilder.com/tool/codon-optimization.html, accessed on 25 July 2022). This webserver was used to produce an improvement in the DNA sequence by codon optimization. After improving the DNA sequence, the sequence was further used for in silico cloning. The webserver provided different results for the given query sequence, including the amount of GC content and the codon adaptation index (CAI) score. For in silico cloning, the pET 30a vector was selected. Moreover, the CAI score can be used to determine the expression levels of the protein [31].

### 2.13. Analysis of Immune Simulation

To perform immune simulation analysis, the C-IMMSIM server 1.0 (https://kraken.iac.rm.cnr.it/C-IMMSIM/, accessed on 28 June 2022) was used. The server performed the immune simulation for the given vaccine. The immune response profile was generated and the immunogenicity of the chimeric peptides was predicted for the given peptide. The webserver (C-IMMSIM) performs the analysis based on Position Specific Scoring Matrices (PSSM). The program is based on machine learning techniques for predicting immune interactions. During the analysis of the vaccine candidate, the default parameters were used.

### 2.14. Protein Docking Analysis

To perform the docking analysis, the TLR2 structure was retrieved from a protein databank. To fetch the TLR2 structure, the Protein Data Bank was used (ID: 2Z7X). To perform the docking analysis, the ClusPro 2.0 (https://cluspro.bu.edu/login.php, accessed on 30 June 2022) webserver was used. The predicted epitopes’ structure was docked against the TLR2 structure and the binding potential was analyzed. The result with the best score was selected and the docking analysis was examined by the Ligplot tool.

### 2.15. Molecular Dynamics Simulation

Molecular dynamics simulation analysis was performed for the docked vaccine and TLR2 complex. The simulation analysis was performed by the iMODs webserver 1.0 (http://imods.chaconlab.org/, accessed on 15 July 2022). The analysis was performed to confirm the stability of the docked compound (receptor and ligand molecules). Hence, the analysis predicts the binding efficiency of the docked molecules, which is very significant in confirming the in silico predictions.

## 3. Results

The retrieved protein sequences of different variants of SARS-CoV-2 related to a surface glycoprotein of the coronaviruses were used to generate a consensus sequence. The consensus sequence is based on 175 sequences of different variants of SARS-CoV-2. Moreover, Geneious Prime software was used to generate the consensus sequence using default parameters. The generated sequence is based on the cost matrix of BLOSUM 62. The length of the consensus sequence is 1273 amino acids.

### 3.1. Prediction of Epitopes and Joining

The generated consensus sequence was used to predict CTL, HTL, and B-cell epitopes. These epitopes are significant in the context of immunity as they induce defensive immunity. The predicted epitopes were filtered and joined after removing redundancy in the epitopes from raw data. Altogether, six B-cell epitopes were selected after removing the noise. Moreover, CTL and HTL epitopes were predicted using the NetCTL 1.2 server (https://www.cbs.dtu.dk/services/NetCTL/, accessed on 5 March 2022). The predicted CTL and HTL epitopes were filtered according to the threshold values. The epitopes were arranged according to their positions and redundancies were removed. Altogether, seven CTL and four HTL epitopes were selected after the filtration of the raw data. In addition, Cholera Toxin B (CTB) adjuvant was added by EAAAK linker to the N-terminal of the candidate vaccine construct as it can induce regulatory immune responses. To make a single peptide, all predicted and selected B- and T-cell epitopes were combined by GPGPG and AAY spacers. A GPGPG spacer was used to join the B-cell epitopes and HTL epitopes, while an AAY linker was introduced to join the CTL epitopes Figure 2.

### 3.2. Enrichment Analysis for Epitopes

The enrichment analysis was performed for combined B- and T-cell epitopes. Altogether, the combined peptide consists of 368 amino acids with a molecular weight of 39.58763 kDa. The theoretical pI of the protein is 8.57, while the total number of negatively charged residues (Asp + Glu) is 27 and the total number of positively charged residues (Arg + Lys) is 31. The atomic composition of the protein highlights that it contains 1804 carbon, 2689 hydrogen, 469 nitrogen, 520 oxygen, and 10 sulfur atoms. The protein consists of a total number of 5492 atoms. Hence, the formula of the protein is C_1804_H_2689_N_469_O_520_S_10_. Moreover, an estimated half-life of the protein is 7.2 h (mammalian reticulocytes, in vitro), more than 20 h (yeast, in vivo), and more than 10 h (*Escherichia coli*, in vivo). The instability index of the protein is computed to be 24.35. Hence, the protein classifies as stable. Furthermore, the aliphatic index of the protein is 65.03 and the grand average of hydropathicity (GRAVY) is −0.401. In the context of trans-membrane helix analysis, no trans-membrane helix was found in the protein. Moreover, the predicted probability of antigenicity of the peptide was 0.930081 as predicted by Antigen pro, while overall protective antigen prediction was 0.5616 as predicted by VaxiJen, which predicted it as an antigen on the basis of the default threshold. The protein was found to be soluble with a probability of 0.849958.

### 3.3. Structure Modeling

The predicted protein structure is shown in Figure 3A. The protein was scanned for quality, wherein initially 84.211% residues were found in the highly preferred region, while 10.877% of residues were found in the preferred region. The quality of the protein was improved, and after improvement, 95.439% residues are found in the highly preferred region, while 3.509% of residues are found in the preferred region. Ramachandran plot analysis of the protein is given in Figure 3B. Moreover, the ProSA-web tool was used to verify the quality of the 3D model. The Z score estimated by the ProSA-webserver was calculated to be −3.07. Furthermore, the overall quality factor calculated by the ERRAT webserver was 71.2803.

### 3.4. Vaccine–Receptor Interaction Visualization Using PDBSum

The docking analysis was performed against the TLR2 structure by using the ClusPro 2.0 webserver. Hence, the receptor protein was docked against the predicted vaccine structure and the binding potential of both proteins was analyzed. The docking analysis was examined by PDBSum, which highlighted interactions of the constructed vaccine against the TLR2 structure. According to the analysis, the protein was interacting at significant points with TLR2 Figure 4.

### 3.5. Codon Optimization and Reverse Translation and in Silico Cloning

Codon optimization procedures were executed using the VectorBuilder webserver, facilitating the analysis and delivery of the refined cDNA sequence. The tool effectively enhanced the DNA sequence, resulting in a revised sequence spanning 1104 nucleotides. Furthermore, the program undertook codon optimization efforts, yielding a codon adaptation index of 0.92 for the improved query pertaining to the candidate vaccine. Additionally, the amplified GC content of the enhanced DNA sequence attained a value of 59.96%. Based on our analytical findings, there is compelling evidence to suggest the proficient expression of the anticipated candidate vaccine within the E. coli K-12 strain. This assertion emanates from the premise that the codon adaptation index is a reliable determinant in this context [31]. The value greater than 0.8 is significant. Moreover, GC content between 30 and 80% is considered significant for good protein expression in the host system [32]. An evaluation using in silico restriction cloning of the reverse translation of the vaccine construct into the pET 30a (+) vector was performed.

### 3.6. Analysis of Immune Simulation

The immune simulation was performed by the C-IMMSIM immune server. This server generates an in silico immune response. Hence, the immunogenic profile of the query candidate vaccine was assessed, as shown in Figure 5. During the simulation, the immune responses were significantly higher. Hence, due to the significant response, the antigenic concentration was decreased while the immunoglobulin activities were increased with the passage of time. These immunoglobulins included IgM and IgG. Moreover, there was a combination of IgM and IgG immunoglobulins. Furthermore, the B-cell population was also high and long-lasting. The results suggest isotype-switching potentials and memory formation, as shown in Figure 5. Similarly, in the context of immunity, the responses of T helper and T cytotoxic cells were also significantly higher, as shown in Figure 5. Hence, based on the computational analysis, the immune response was significantly generated at high levels during the process of immune stimulation.

### 3.7. Molecular Dynamics Simulation

The molecular dynamic simulation result from the iMOD webserver of the vaccine construct and the TLR2 complex is shown in Figure 6. The simulation analysis highlighted the movement of atoms within a rigid body of the vaccine construct. In the figure, the main chain deformity graph of the docked structure is highlighted. The peaks of the graph are indicative of the distortions within the protein regions. The B-factor graph seen in Figure 6B shows the relationship between the NMA and the corresponding PDB field. Moreover, Figure 6C predicts the eigenvalue of the construct to be 8.666098 × 10^−6^ related to the motion of the structure. The inverse of the eigenvalue is the variance of the structure, which is described in Figure 6D. The graph shows the cumulative variance in green color and the individual variance in red color. Another factor calculated by this online tool is a covariance matrix, which indicates coupling between pairs of residues. The residues could either be correlated, uncorrelated, or anti-correlated in motion, as seen in Figure 6E. The last prediction is an elastic network map of the docked complex. Each dot in the graph represents one spring between the corresponding pair of atoms. The stiffness of the atoms is indicated by a grey color range in Figure 6F. Darker grey dots predict stiffer areas in residues, whereas lighter grey dots predict flexible ones. The molecular simulation conducted by the IMODs server suggests that the docked vaccine construct with the TLR2 complex is stable and, therefore, further analysis can proceed.

## 4. Discussion

In the last two decades, severe acute respiratory syndrome coronavirus (SARS-CoV), Middle East respiratory syndrome coronavirus (MERS-CoV), and severe acute respiratory syndrome coronavirus-2 (SARS-CoV-2) are three pathogenic and highly transmissible viruses that emerged in humans. These coronaviruses originated in bats and were then transferred to humans using some intermediate hosts. For example, civets served as an intermediate host for SARS-CoV, camelids as intermediate hosts for MERS-CoV, and, probably, pangolins served as an intermediate host for SARS-CoV-2 [33,34]. Coronaviruses undergo genetic recombination and evolution that may result in new recombinant coronaviruses that may be more deadly to humans [35]. The S gene, which encodes spike protein, which contains a receptor-binding domain, is the most vulnerable for recombination [36,37]. Looking at the prevalence and genetic diversity, along with the frequent genetic recombination, of the coronaviruses, it is expected that novel variants will emerge in the future, which may lead to SARS-CoV-2-like pandemics, or even more deadly ones [38].

The neutralizing antibody’s responses and T-cell responses can be enhanced against many coronavirus proteins, but they mainly target spike protein (S), suggesting that S protein plays an important role in fighting against coronavirus infections [39,40]. The coronavirus S protein involves virus attachment and entry in the host cell, which makes the S protein the most important target for therapeutics and vaccine development [6]. It is possible to develop a single vaccine by using spike protein that will be able to protect against SARS-CoV-2 variants infections. Recently, it was discovered that SARS-CoV S protein induces a polyclonal antibody response and strongly neutralizes SARS-CoV-2 S protein-mediated entry into cells. Therefore, the prospect of using spike protein as a target for coronavirus vaccination and immunotherapy is encouraging [41]. Further pieces of evidence suggested that ARS-CoV RBD-specific antibodies cross-react with SARS-CoV-2 RBD protein, and SARS-CoV RBD induced antisera to neutralize SARS-CoV-2, which provides additional evidence to suggest that we target the S protein of SARS-CoV-2 for a vaccine that could be effective in preventing SARS-CoV-2 [42].

This work, therefore, focused on the in silico designing and development of a potential multi-epitope peptide vaccine that will be able to protect against multiple types of coronaviruses. For this purpose, spike protein sequences of 175 SARS-CoV-2 variants were selected and the consensus sequence was developed. This consensus sequence was utilized to predict T-cell and B-cell epitopes. The assessments of T-cell responses are very important for immunity and protection against coronavirus infections. SARS-CoV-2-specific memory CD8 T cells can protect susceptible hosts from lethal SARS-CoV-2 infection, but they also indicate that SARS-CoV-2-specific CD4 T cells and antibody responses are necessary for comprehensive protection. Thus, CD8 and CD4 T cells are necessary for complete protection against coronaviruses [15]. The B-cell, CTL, and HTL epitopes were predicted using online webservers. The predicted epitopes were filtered on the basis of the threshold value [43]. Moreover, the epitopes were further filtered on the basis of toxicity, antigenicity, and allergenicity analysis. Finally, seven CTL, four HTL, and six B-cell epitopes were selected after removing redundancies, and then these epitopes were arranged according to their positions. The B cells are important for antibody production against the vaccinated antigen. During coronavirus infection, local immune responses release cytokines and prime adaptive T-cell and B-cell responses. Thus, B-cell response is very important to provide long-lasting immunity against coronavirus infections [44]. The potential B-cell epitopes were predicted using bioinformatics software, and the top five were selected for the novel candidate vaccine development.

Based on their allergenicity, antigenicity, and conservational analysis, the top CTL, HTL and B-cell epitopes were selected and combined using the linkers to make a single multi-epitope peptide. To link these epitopes, the GPGPG and AAY linkers were used, which improves the immune processing of the antigen and also prevents the junctional epitope formation. The glycine-rich linker (GPGPG) and AAY were preferred to join the candidate vaccine epitopes because their use improves the solubility and makes the vaccine domains accessible and active [43]. In addition, Cholera Toxin B (CTB) adjuvant was added by EAAAK linker to the N-terminal of the vaccine construct as it can induce regulatory immune responses [45]. The designed multi-epitope candidate vaccine was predicted to be non-allergenic, having the good physicochemical properties of a good candidate vaccine against coronaviruses [24,46]. The molecular weight of the candidate vaccine construct was 39.587 kDa and the instability index classifies the candidate vaccine as stable. The GRAVY index of the vaccine was 0.0401, which predicts that the vaccine is polar and has a good capability of water interaction [47]. The aliphatic index indicated that the vaccine is thermostable, which is a quality of a good vaccine [48]. The half-life of the predicted candidate vaccine was 7.2 h (mammalian reticulocytes, in vitro); this indicates the time used by the protein to reach 50% concentration after its synthesis in the cell [49]. The structure validation of the vaccine construct was performed using Ramachandran plot analysis. The score validates the overall quality of the vaccine construct [50]. Furthermore, the predicted vaccine was docked with a TLR2 structure. The high docking score predicts the better stimulation of immune responses to this vaccine.

To check an effective vaccine expression in *Escherichia coli* cells, codon optimization was performed, and the peptide was reverse-translated into specific cDNA. The GC content of the vaccine was 59.96%, which shows good expression of the vaccine peptide into a prokaryotic host. The in silico cloning of the vaccine was performed, using expression vector pET-30a (+) for its expression, in a prokaryotic expression system. The in silico studies of the immune simulation confirmed that the designed candidate vaccine was able to stimulate the good immune responses that are required for a good vaccine. Based on the docking score, immunogenicity score, population coverage, and physicochemical properties, the predicted candidate vaccine may prove highly immunogenic against coronaviruses.

## 5. Conclusions

We developed an in silico-based universal candidate vaccine against coronaviruses. The computational analysis revealed that the designed candidate vaccine is stable and non-allergenic in the human body. Moreover, the efficiency of the multi-epitope candidate vaccine to trigger effective cellular and humoral immune responses was checked by immune stimulation, which revealed that the newly designed candidate vaccine can trigger specific immune responses against multiple coronaviruses, and thus, it can be a potential candidate vaccine against existing and coming coronaviruses.

## Figures and Tables

**Figure 1 microorganisms-11-02282-f001:**
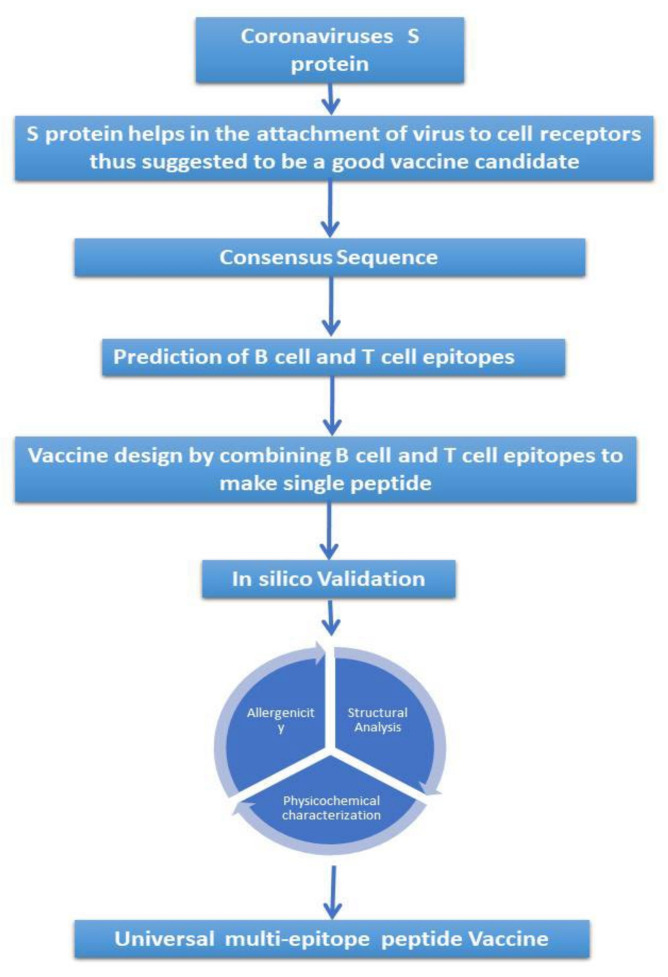
Flowchart diagram of methods used for the prediction of a vaccine. The diagram explains the sequence of methods used for the prediction of a universal multi-epitope vaccine against SARS-CoV-2 and its variants.

**Figure 2 microorganisms-11-02282-f002:**
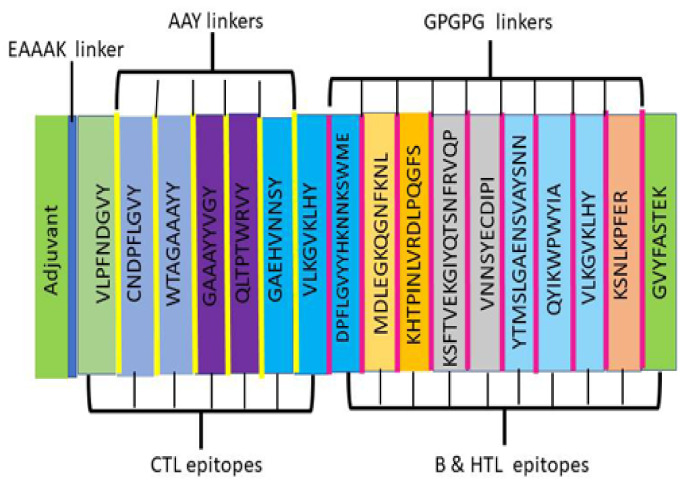
Generation of multi-epitope vaccine construct. Geneious Prime software was used to generate the consensus sequence. The predicted epitopes and adjuvant are joined to make a single peptide. The graphics of the vaccine construct are shown.

**Figure 3 microorganisms-11-02282-f003:**
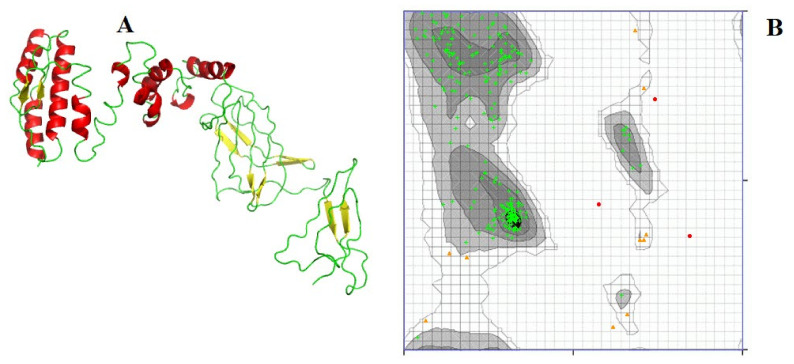
The predicted protein structure of the vaccine construct and its quality are shown in the figure. The structure was predicted by threading algorithm using the I-Tasser webserver. (**A**) Red color shows alpha helices, while yellow color shows beta sheets in protein structure. Green color shows turns and loops. The protein was scanned for quality, wherein initially 84.211% of residues were found in the highly preferred region, while 10.877% of residues were found in the preferred region. The quality of the protein was improved, and after improvement, 95.439% of residues are found in the highly preferred region, while 3.509% of residues are found in the preferred region. (**B**) Plotting the φ values on the x-axis and the ψ values on the y-axis to predict the possible conformation of the peptide. The graph shows different favorable areas for amino acids. On the basis of these areas, the quality of the protein structure is determined. The green crosses donate different amino acids. Red: allowed regions of the ϕ and ψ dihedral angle space; Yellow triangle: disallowed or sterically unfavorable regions of the ϕ and ψ dihedral angle space.

**Figure 4 microorganisms-11-02282-f004:**
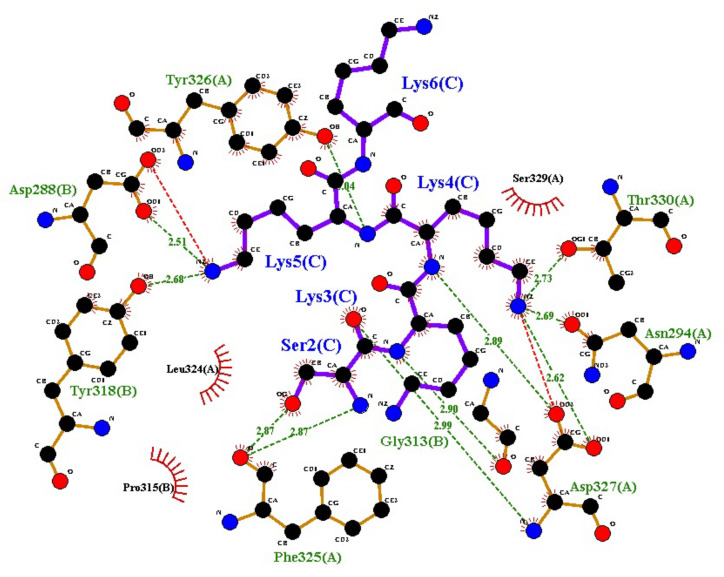
Schematic diagram of interaction between constructed vaccine and TLR2 human receptor. Chain A is constructed vaccine and B is TLR2 receptor. Residue colors: positive (H,K,R); negative (D,E); S,T,N,Q = neutral; A,V,L,I,M = aliphatic; F,Y,W = aromatic; P,G = Pro and Gly; C = cysteine. Chain A represents the vaccine and B TLR2. No. of interacting residues from chain A was 25, and 27 residues made interactions from chain B. The interface area from chain A was 1481 Angstrom and from chain B it was 1431 Angstrom. There were 8 hydrogen bonds and, in total, 167 non-bonded interactions between both proteins.

**Figure 5 microorganisms-11-02282-f005:**
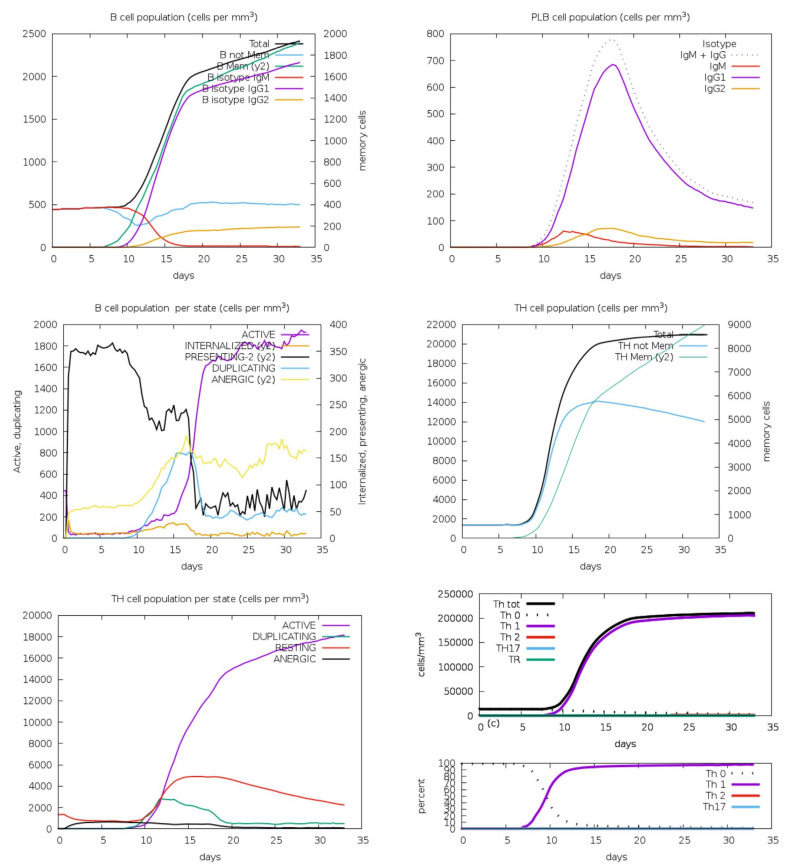
The immune simulation results. The simulation was performed by the C-IMMSIM immune server. During the simulation, the immune responses were significantly higher. Hence, B-cell isotypes were also found which were long-lasting. The results suggest isotype-switching potentials and memory formation.

**Figure 6 microorganisms-11-02282-f006:**
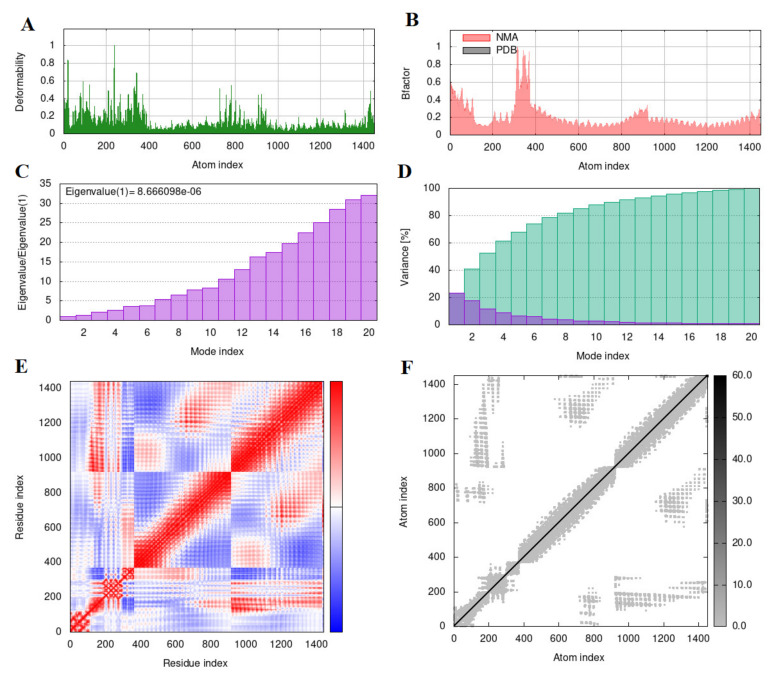
The molecular dynamic simulation results of the vaccine construct and the TLR2 complex. The analysis was performed using the iMOD webserver. (**A**) The B-factor graph shows the relationship between the NMA and the corresponding PDB field. (**B**) The eigenvalue of the construct is 8.666098 × 10^−6^ related to the motion of the structure. (**C**).The inverse of the eigenvalue is the variance of the structure is shown in (**D**). Residues could either be correlated, uncorrelated, or anti-correlated in motion, seen in (**E**). The stiffness of the atoms is indicated by a grey color range in (**F**).

## Data Availability

The research data will be available upon request to corresponding author.

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
