# Peer review of "Design and Assessment of a Novel In Silico Approach for Developing a Next-Generation Multi-Epitope Universal Vaccine Targeting Coronaviruses"

_microorganisms, 2023, doi:10.3390/microorganisms11092282_

Round 1
Reviewer 1 Report
I cannot fully evaluate this manuscript without Figure 2. My sense is that with a lot of work and polishing, the authors have made a contribution that can be published, and that it is appropriate to publish that eventual article in this or another MDPI journal.
"One advantage is that coronaviruses have a relatively low mutation rate compared to other RNA viruses, which can be utilized to develop a multi-epitope vaccine capable of providing protection against current and future coronaviruses"
This is not true. Coronaviruses mutate quickly. The fast mutation rate caught the world off guard and led to a large number of variants of concern, such as omicron.
"In this study, we employed novel computational approaches to analyze genetically related coronaviruses, including SARS-CoV-2 and its"
The word 'novel' does not belong in a title or abstract. Just say exactly what you did and let the readers figure out whether it is novel or not
"In-silico codon optimization and cloning further confirmed successful expression of the designed candidate vaccine in a prokaryotic expression system"
In silico cannot confirm something in vivo. Maybe you mean 'suggest' or 'Explore' rather than 'confirm'.
Explain why just TLR2 was used, and not other TLRs. More generally, explain why TLRs were examined and not TCRs or other antigen receptors.
Figure 3.
A. legend need to explain the meaning of the colors
B. Axes need labeling. Meaning of the contour lines and colors and shape of icons needs explanation.
Run entire manuscript through a spelling and grammar checker to avoid spelling errors such as "preffered". Google Docs offers a free grammar and spelling checker. And weird grammar like: "The of coronavirus (SARS-CoV-2) was originated in Wuhan". And bad punctuation like "human cell.. This"
Consider deleting your entire first paragraph. It has awful grammar. Everyone reading the article already knows all this. They can find it in Wikipedia.
Figure 5. I am not sure Figure 5 is needed. the entire article is about speculative in silico analyses. If anyone (e.g., a vaccine company) were to actually make this vaccine, they would surely design their own construct. I suppose you are showing that a construct in principle can be done. Maybe put this in Supplemental. At the very least, delete all the annotations of restriction sites. They clutter up the diagram. Way too much white space.
Figure 6. Make sure all axes are labeled, including with a specification of the units. Increase the resolution of the figures. You can actually see poor cropping a low resolultion multifigure as the top of some text appears at the bottom.
Figure 2 is missing.
Introduction:
consider citing recent articles about computational identification of universal coronavirus vaccines. I found nine hits in Pubmed to this search:
https://pubmed.ncbi.nlm.nih.gov/?term=%22universal+coronavirus+vaccine%22&sort=date
"universal coronavirus vaccine"
Explain why your article is different or builds on each of the relevant refernces you identify
Consider one or more of these:
Gustiananda M, Julietta V, Hermawan A, Febriana GG, Hermantara R, Kristiani L, Sidhartha E, Sutejo R, Agustriawan D, Andarini S, Parikesit AA. Immunoinformatics Identification of the Conserved and Cross-Reactive T-Cell Epitopes of SARS-CoV-2 with Human Common Cold Coronaviruses, SARS-CoV, MERS-CoV and Live Attenuated Vaccines Presented by HLA Alleles of Indonesian Population. Viruses. 2022 Oct 24;14(11):2328. doi: 10.3390/v14112328. PMID: 36366426; PMCID: PMC9699331.
Lim CP, Kok BH, Lim HT, Chuah C, Abdul Rahman B, Abdul Majeed AB, Wykes M, Leow CH, Leow CY. Recent trends in next generation immunoinformatics harnessed for universal coronavirus vaccine design. Pathog Glob Health. 2023 Mar;117(2):134-151. doi: 10.1080/20477724.2022.2072456. Epub 2022 May 12. PMID: 35550001; PMCID: PMC9970233.
Vashishtha VM, Kumar P. Looking to the future: is a universal coronavirus vaccine feasible? Expert Rev Vaccines. 2022 Mar;21(3):277-280. doi: 10.1080/14760584.2022.2020107. Epub 2021 Dec 30. PMID: 34968153.
Koff WC, Berkley SF. A universal coronavirus vaccine. Science. 2021 Feb 19;371(6531):759. doi: 10.1126/science.abh0447. PMID: 33602830.
Giurgea LT, Han A, Memoli MJ. Universal coronavirus vaccines: the time to start is now. NPJ Vaccines. 2020 May 28;5(1):43. doi: 10.1038/s41541-020-0198-1. PMID: 32528732; PMCID: PMC7256035.
Grammar is atrocious. The authors need to carefully read the final formatted draft of their manuscript before submitting it.
Author Response
- Comment:
I cannot fully evaluate this manuscript without Figure 2. My sense is that with a lot of work and polishing, the authors have made a contribution that can be published, and that it is appropriate to publish that eventual article in this or another MDPI journal.
Author Response:
As per suggestion, the figure 2 is added in the revised manuscript along with its labels.
- Comment:
"One advantage is that coronaviruses have a relatively low mutation rate compared to other RNA viruses, which can be utilized to develop a multi-epitope vaccine capable of providing protection against current and future coronaviruses".
This is not true. Coronaviruses mutate quickly. The fast mutation rate caught the world off guard and led to a large number of variants of concern, such as omicron.
Author Response:
As per recommendation, we have removed the statement from the abstract and updated in revised manuscript.
- Comment:
"In this study, we employed novel computational approaches to analyze genetically related coronaviruses, including SARS-CoV-2 and its"
The word 'novel' does not belong in a title or abstract. Just say exactly what you did and let the readers figure out whether it is novel or not
Author Response:
As per suggestion, we have removed the world novel from the abstract and updated in revised manuscript.
- Comment:
"In-silico codon optimization and cloning further confirmed successful expression of the designed candidate vaccine in a prokaryotic expression system"
In silico cannot confirm something in vivo. Maybe you mean 'suggest' or 'Explore' rather than 'confirm'.
Author Response:
As per recommendation, the “confirmed” word is replaced with “explore” in the abstract of updated manuscript.
- Comment:
Explain why just TLR2 was used, and not other TLRs. More generally, explain why TLRs were examined and not TCRs or other antigen receptors.
Author Response:
TLR-2 receptors have a vital function in vaccine development and manufacturing by participating in the immune response by identifying pathogen-associated molecular patterns (PAMPs) and TLR-2 is primarily involves in the recognition of viral proteins which leads to particular immune response and complements the functioning of other TLRs for better immune response.
TLRs provide a rapid and general response to a broad range of pathogens, initiating the first line of defense while TCRs are capable of generating specific response related to that’s why we have targeted TLRs in our study.
- Comment:
Figure 3. (A). legend need to explain the meaning of the colors
- Axes need labeling. Meaning of the contour lines and colors and shape of icons needs explanation.
Author Response:
As per suggestion, The legend and color meanings are added for Figure 3. (A) and for (B) and updated in the revised manuscript.
- Comment:
Consider deleting your entire first paragraph. It has awful grammar. Everyone reading the article already knows all this. They can find it in Wikipedia.
Author Response:
As per suggestion, we have rephrased the entire first paragraph and rectify the grammar and spelling mistakes in the revised manuscript.
- Comment:
Figure 5. I am not sure Figure 5 is needed. the entire article is about speculative in silico analyses. If anyone (e.g., a vaccine company) were to actually make this vaccine, they would surely design their own construct. I suppose you are showing that a construct in principle can be done. Maybe put this in Supplemental. At the very least, delete all the annotations of restriction sites. They clutter up the diagram. Way too much white space.
Author Response:
As per recommendation, we have removed the Figure 5. in the updated manuscript.
- Comment:
Figure 6. Make sure all axes are labeled, including with a specification of the units. Increase the resolution of the figures. You can actually see poor cropping a low resolultion multifigure as the top of some text appears at the bottom.
Author Response:
We have replaced the Figure 5. with high resolution image as per suggestion in the updated manuscript.
- Comment:
Figure 2 is missing.
Author Response:
As per suggestion, the figure 2 is added in the revised manuscript along with its labels.
- Comment:
consider citing recent articles about computational identification of universal coronavirus vaccines. I found nine hits in Pubmed to this search:https://pubmed.ncbi.nlm.nih.gov/?term=%22universal+coronavirus+vaccine%22&sort=date
"universal coronavirus vaccine"
Explain why your article is different or builds on each of the relevant refernces you identify
Author Response:
This study stands out by taking a holistic approach that encompasses a wider range of coronaviruses, employs advanced computational techniques, and validates its findings through a comprehensive series of in-silico analyses and immunogenicity assessments. It offers a promising avenue for the development of a multi-epitope vaccine capable of combatting various coronaviruses, making it a valuable contribution to the field.
Reviewer 2 Report
This manuscript identified the top epitopes and created a multi-epitope candidate vaccine. However, the results are not convincing and solid. For most of the results, the authors did not provide enough analysis. There are a lot of issues in writing. This manuscript cannot be published in Microorganisms. The main issues are listed as follows:
1. There is no Figure 2. Please add the figure.
2. The description of figure 3B is the same as the content in section 3.3. Please rewrite this description for 3B and clearly indicate what the meaning of each label is in the figure. For example, what does the red dot represent for?
3. Please add more details for section 3.4. What kinds of points are significant points which are mentioned in section 3.4? And, how the protein interacts with TLR structure? Please describe those interactions.
4. Figure 4 is not clear enough. Please rewrite the legend of figure 4. Please don’t copy and paste the content from results. The authors should describe the figure in the figure legend. For example, what is the meaning of each colored dash line between residues?
5. Please add references for this sentence “GC content between 30 and 80% is considered significant for good protein expression in the host system” in section 3.5.
6. Please add more analysis for Figure 6. Figure 6 is not clear enough. Please make it clearer. And name each chart in the figure legend. Analyze each chart in section 3.6.
7. At the end of section 3.7, the authors said, “The molecular simulation conducted by the IMODs server suggests that the docked vaccine construct with TLR2 complex is stable and can therefore proceeded for further analysis.” But there is no analysis of each chart. How did the authors draw this conclusion? Please add more analysis.
The quality of English language is fine.
Author Response
- Comment:
There is no Figure 2. Please add the figure.
Author Response:
As per suggestion, the figure 2 is added in the revised manuscript along with its labels.
- Comment:
The description of figure 3B is the same as the content in section 3.3. Please rewrite this description for 3B and clearly indicate what the meaning of each label is in the figure. For example, what does the red dot represent for?
Author Response:
This suggestion is already incorporated in the updated manuscript.
- Comment:
Please add more details for section 3.4. What kinds of points are significant points which are mentioned in section 3.4? And, how the protein interacts with TLR structure? Please describe those interactions.
Author Response:
As per suggestion, we have incorporated the details in the revised manuscript.
- Comment:
Figure 4 is not clear enough. Please rewrite the legend of figure 4. Please don’t copy and paste the content from results. The authors should describe the figure in the figure legend. For example, what is the meaning of each colored dash line between residues?
Author Response:
As per suggestion, figures legends are incorporated in the revised manuscript.
- Comment:
Please add references for this sentence “GC content between 30 and 80% is considered significant for good protein expression in the host system” in section 3.5.
Author Response:
As per your suggestion Reference number 32 has been added
- Comment:
Please add more analysis for Figure 6. Figure 6 is not clear enough. Please make it clearer. And name each chart in the figure legend. Analyze each chart in section 3.6.
Author Response:
Good quality Figure 6 is required
Author Response: Good quality figure has been added
- Comment:
At the end of section 3.7, the authors said, “The molecular simulation conducted by the IMODs server suggests that the docked vaccine construct with TLR2 complex is stable and can therefore proceeded for further analysis.” But there is no analysis of each chart. How did the authors draw this conclusion? Please add more analysis.
Author Response: All the suggestions and changes has been made in the manuscript
Reviewer 3 Report
It could be an interesting documment related to vaccines development, but it is a very technical manuscript that I think it will be difficult to understand by people that are not experts in computacional work (I have had problems to understand the different parts of the methodology and the results.
According to the idea that it could be very interesting to understand the dinamics of the vaccines design and the obstacles in the case of coronavirus vaccines I considere it could be better to present it in a less technical level
Attentio toa single comment in the discussion.

Author Response
All the suggested changes have been made in the revised manuscript
Round 2
Reviewer 1 Report
Please add the text in the responses (from Round 1) to the reviewer to the paper. If the reviewer is interested in a question, some of the readers will be as well.
Figure 2: Any sequence data needs to be copy/pastable and machine readable, so it can not be included as an image file. You could include it as supplemental material in a text file, or in a table.
Author Response
Figure 2: Any sequence data needs to be copy/pastable and machine readable, so it can not be included as an image file. You could include it as supplemental material in a text file, or in a table.
Reply: Thanks for your suggestion. The figure 02 has been modified according to your suggestion.
Reviewer 2 Report
At the end of section 3.7, the authors said, “The molecular simulation conducted by the IMODs server suggests that the docked vaccine construct with TLR2 complex is stable and can therefore proceeded for further analysis.” The authors just described each chart. But they did not add any analysis. If there is no analysis, how did the authors draw this conclusion? Please add more analysis for these results.
The quality of English language is fine.
Author Response
At the end of section 3.7, the authors said, “The molecular simulation conducted by the IMODs server suggests that the docked vaccine construct with TLR2 complex is stable and can therefore proceeded for further analysis.” The authors just described each chart. But they did not add any analysis. If there is no analysis, how did the authors draw this conclusion? Please add more analysis for these results.
Reply: Thank you very much for your suggestion. This study was mainly focused on the insilico prediction of the Coronavirus vaccine. The "further analysis" mentioned in your comment means the invivo studies.